# Real-Time PCR-Based Test as a Research Tool for the Retrospective Detection and Identification of SARS-CoV-2 Variants of Concern in a Sample

**DOI:** 10.3390/ijms26051786

**Published:** 2025-02-20

**Authors:** Valeria O. Makarova, Artem Shelkov, Anna Iliukhina, Valentin Azizyan, Inna V. Dolzhikova, Elena Vasilieva, Alexey A. Komissarov

**Affiliations:** 1Moscow City Clinical Hospital Named after I.V. Davydovsky, 109240 Moscow, Russia; leroook.mak@gmail.com (V.O.M.); vasilievahelena@gmail.com (E.V.); 2Federal State Budget Institution “National Research Centre for Epidemiology and Microbiology Named After Honorary Academician N F Gamaleya” of the Ministry of Health of the Russian Federation, 123098 Moscow, Russia; artem.shelkov@gmail.com (A.S.); sovanya97@yandex.ru (A.I.); valentin.www-mark@mail.ru (V.A.); iv.dolzhikova@yandex.ru (I.V.D.); 3FSBEI HE “Russian University of Medicine” of the Ministry of Health of the Russian Federation, 127473 Moscow, Russia; 4National Research Centre “Kurchatov Institute”, 123182 Moscow, Russia

**Keywords:** SARS-CoV-2, COVID-19, variants of concern, real-time PCR

## Abstract

The Severe Acute Respiratory Syndrome-related Coronavirus 2 (SARS-CoV-2), a causative agent of the COVID-19 disease, has been constantly evolving since its first identification. Mutations that are embedded in the viral genomic RNA affect the properties of the virus and lead to the emergence of new variants. During the COVID-19 pandemic, the World Health Organization has identified more than ten variants of the SARS-CoV-2 virus. Five of these—Alpha, Beta, Gamma, Delta, and Omicron—were classified as variants of concern (VOCs), as they caused significant outbreaks of the disease. Additionally, two progeny variants of Omicron, designated JN.1 and KS.1, are still causing new waves of infections. Due to the emergence of various SARS-CoV-2 variants, in some cases, it has become important to identify a particular variant in a sample. Here, we have developed a multiplexed probe-based real-time PCR system for the identification of SARS-CoV-2 VOCs (Alpha, Beta, Gamma, Delta, Omicron B.1.1.529/BA.1, and Omicron BA.2), as well as modern Omicron variants JN.1 and KS.1. The sensitivity and specificity of the PCR system have been tested using isolated viral genomes and RNA preparations from human nasopharyngeal swabs. The system allows for rapid identification of coronavirus variants in the cryopreserved and fresh samples.

## 1. Introduction

In December 2019, a new virus from the Betacoronavirus family, Severe Acute Respiratory Syndrome-related Coronavirus 2 (SARS-CoV-2), was first identified. Since then, it has spread worldwide, causing the dangerous infectious disease COVID-19. The virus genome, which consists of a single-stranded positive RNA molecule, is constantly changing, affecting the properties of the virus and leading to the emergence of new mutant variants. These mutations, embedded in the genomes of the new variants, have a positive impact on the viral life cycle, allowing for better penetration into host cells and evasion of pre-existing immunity. This makes the virus more efficiently transmitted between humans [1,2]. During the COVID-19 pandemic, the World Health Organization (WHO) has identified more than ten different variants of the SARS-CoV-2 virus. Five of these variants have caused significant outbreaks of the disease and have been classified as variants of concern (VOCs): Alpha, Beta, Gamma, Delta, and Omicron. Although there are currently no SARS-CoV-2 variants that meet the criteria for VOC status, two progeny variants of Omicron, known as JN.1 and KS.1, are still causing new waves of infections [3]. Due to the emergence of numerous variants of the virus, it has become essential to not only detect the presence of coronavirus in samples but also identify the specific variant in some cases.

The most effective method for detecting a specific SARS-CoV-2 variant is high-throughput RNA sequencing, as it allows for the determination of the genomic RNA sequence and the entire spectrum of accumulated mutations. However, this technique is time-consuming and costly. An alternative to RNA sequencing is real-time polymerase chain reaction with reverse transcription (RT-PCR), as this method makes it easier and less expensive to identify a viral variant. However, using this method requires access to sensitive and specific PCR sets that are oligonucleotides serving as primers and probes. To date, PCR sets that can efficiently detect variants of SARS-CoV-2 have been developed [4,5,6,7,8,9,10,11]. However, these sets are based on the detection of the same mutations in the S-protein gene single-nucleotide substitutions (K417N/T, L452R, T478K, E484K, N501Y, and P681R/H), as well as the H69/V70 deletion. Meanwhile, we have not found any published literature on PCR sets specific to the JN.1 or KS.1 sublineages of the Omicron variant.

In this study, we have developed a probe-based multiplexed PCR system for the simultaneous detection of five SARS-CoV-2 VOCs based on alternative mutations. The system also allows for the detection of modern SARS-CoV-2 variants (Omicron sublineages JN.1 and KS.1), as well as human RNA as an internal control. We analyzed the available genome sequences of Alpha, Beta, Gamma, Delta, and Omicron variants and identified variant-specific mutations. We designed PCR sets consisting of forward/reverse primers and fluorescent probes to detect these mutations. The sensitivity and specificity of these PCR sets were tested using isolated viral genomes and total RNA preparations from nasopharyngeal swabs of humans infected with various SARS-CoV-2 variants. Finally, we combined the sets in the multiplexed PCR system, which allows the detection of six VOC variants of the SARS-CoV-2 virus (Alpha, Beta, Gamma, Delta, Omicron B.1.1.529/BA.1, and Omicron BA.2), as well as two modern Omicron BA.2 progeny variants, JN.1 and KS.1, through the assessment of three PCR reactions.

## 2. Results

### 2.1. Experimental Testing of the Created PCR Sets of Oligonucleotides

The specificity and sensitivity of the developed PCR sets were analyzed. To assess the sensitivity, standard solutions with serial tenfold dilutions of viral genomes from 10^6^ to 10^2^ copies per PCR reaction were prepared. Using these solutions, a lower detection limit of 10^3^ copies of genomic RNA per reaction was determined for each set since we detected no signal for 102 copies of genomic RNA per reaction within 42 PCR cycles (Figure 1). Next, each set was further tested using the genomes of the variants under study, as well as the wild-type Wuhan genome, and it was shown that the sets effectively amplified only the corresponding genomes, indicating their specificity (Figure 2). However, when analyzing the genome sequences of Omicron BA.2 sublineages JN.1 and KS.1, we noticed that a part of the genomes of these variants contains the same mutation as the parental Omicron BA.2 variant. As a result, in some cases, Omicron JN.1 or Omicron KS.1 variants can be positive for both singleplex PCR sets—Omicron BA.2 and Omicron JN.1-KS.1.

Using the OligoAnalyzer software, we evaluated the compatibility of the developed PCR sets. In addition, to eliminate the possibility of obtaining false negative results, we included a PCR set that amplifies the mRNA of the human ubiquitin C (UBC) gene. This gene has already been shown to have stable expression across different tissues in humans, making it a good choice as a reference [12,13]. The PCR set for the human *UBC* gene was developed previously [14] and, in the current study, was used as an internal control, as it should produce a signal even in SARS-CoV-2-negative samples, indicating the absence of RNA degradation. As a result, we combined the PCR sets into three multiplexes based on their specificity: “ABD”—Alpha, Beta, and Delta; “GOm2JNKS”—Gamma, Omicron BA.2, and Omicron sublineages JN.1 and KS.1; “Om1UBC”—Omicron B.1.1.529/BA.1 and *UBC* gene. Similar to the singleplexes, each multiplex was tested for its sensitivity and ability to specifically amplify the genomes of the corresponding variants (Figure 3).

Thus, we found that there was no significant decrease in the sensitivity of the multiplexed PCR system, and we demonstrated that all multiplexed reactions effectively amplify only the desired genomes, indicating their specificity.

### 2.2. Testing of the Multiplexed PCR System on Human Nasopharyngeal Swabs

Finally, the multiplexed PCR system developed in the study was tested using human samples. For this purpose, we used total RNA previously extracted from nasopharyngeal swabs from healthy donors and infected individuals from the I.V. Davydovsky Moscow City Hospital (see Section 4 for details). In total, 52 samples were tested using the developed multiplexed PCR system in duplicates, i.e., by six PCR reactions (Table 1 and Appendix A). We found that no non-specific amplification was detected in the 11 samples from non-infected patients (Figure 4A). Meanwhile, among the 41 samples from infected individuals, we found that 5 (12%) were infected with the Delta variant (Figure 4B), 5 (12%) with the Omicron B.1.1.529/BA.1 variant (Figure 4C), 19 (46%) with the Omicron BA.2 variant (Figure 4D), and 5 (12%) have been tested positive for Omicron BA.2 progeny variants JN.1/KS.1 (Figure 4E). Unexpectedly, in 7 (17%) cryopreserved RNA samples, we found amplification of two PCR sets—for Gamma and Omicron B.1.1.529/BA.1 variants (Figure 4F).

## 3. Discussion

More than five years have passed since the first detection of SARS-CoV-2 in December 2019. At the early stages of the pandemic, when specific therapeutics and vaccines had not yet been developed, the main concern was to create in vitro diagnostic (IVD) tools to rapidly detect COVID-19 to limit its spread by identifying cases, isolating infected individuals, and tracing their contacts. Among the available IVD methods for detecting SARS-CoV-2 infection, nucleic acid testing has become the primary method since it allows the detection of the presence of the coronavirus earlier than antibody-based methods and is more accurate than clinical imaging techniques (discussed in detail in [15]). Nevertheless, the virus constantly evolves, and its genomic RNA accumulates mutations that upgrade the transmissibility and virulence and therefore give rise to the emergence of new SARS-CoV-2 variants. This poses a challenge for the development of nucleic acid-based IVD assays, which have to be constantly updated and validated to be able to detect new emerging variants [16]. However, since co-infection with different virus variants is a highly rare event [17], as well as emerging SARS-CoV-2 variants tend to replace previous ones from the population [18], IVD tests are commonly designed to detect only those variants that currently dominate worldwide.

In contrast to IVD tests, SARS-CoV-2 detection assays for research purposes should not only detect the presence of viral RNA in a sample but often be able to identify a specific variant. In the case of retrospective analysis, even those variants that are not currently circulating. In this context, next-generation RNA sequencing techniques can be successfully used [19], but these methods are rather expensive, time-consuming, and require sophisticated statistical analysis of the data. Real-time PCR with reverse transcription has become the gold standard for detecting SARS-CoV-2 infections in nasopharyngeal swabs and other samples due to its high specificity and sensitivity, as well as its low cost compared to RNA sequencing methods [20]. Indeed, real-time PCR systems capable of identifying SARS-CoV-2 variants have been previously developed [4,5,6,7,8,9,10,11], although they are similar in nature and rely on the detection of identical mutations in the S-protein gene. Meanwhile, we have not found any published literature on PCR systems specific to the JN.1 or KS.1 sublineages of the Omicron variant, probably because these variants have recently emerged.

In this study, we aimed to develop a multiplexed PCR system that can rapidly and accurately identify five key SARS-CoV-2 variants of concern and modern variants. For this purpose, we first performed in silico analysis of the viral genomes and identified mutations that are specific to certain variants. These mutations include deletions and multiple nucleotide substitutions and have been described in the previous literature. Given their nature, it is unlikely that these mutations will be reversed accidentally, unlike single-nucleotide substitutions used in previous PCR systems. We have designed oligonucleotide primers and probes to specifically amplify the identified variant-specific mutations and showed their high specificity and sensitivity using purified genomic RNA of various SARS-CoV-2 variants both in single and multiplexed formats. Additionally, we tested the performance of the multiplexed PCR system on total RNA isolated from human nasopharyngeal swabs and proved its efficiency by identifying different SARS-CoV-2 variants in the samples. It is important to note that the frequencies of the coronavirus variants detected do not reflect the actual distribution of viral variants in the population. This is because the RNA samples from the cryobank were collected randomly. Nevertheless, all the coronavirus VOC variants detected (Delta, Omicron B.1.1.529/BA.1, and Omicron BA.2), as well as Omicron JN.1-KS.1, were indeed represented in the Moscow population during the period of sample collection [21].

In our view, the most intriguing finding is that we found six RNA samples among the analyzed ones in which effective amplification was observed by two PCR oligonucleotide sets—for Gamma and Omicron B.1.1.529/BA.1 variants. These results could indicate either a simultaneous infection with both VOC variants or an infection with a viral variant containing both variant-specific mutations. The first option seems unlikely since the infection peaks for Gamma and Omicron B.1.1.529/BA.1 are spaced in time by approximately 5 months, and the Gamma variant has never been abundant in Moscow [21]. Meanwhile, the reported co-infections were established between SARS-CoV-2 variants whose temporal distributions overlap, e.g., Gamma and Delta or Delta and Omicron BA.1 [17]. There is still a possibility that several Moscow residents returned from a foreign country, were infected with the Gamma variant, and later became infected with the Omicron B.1.1.529/BA.1 variant in Moscow. However, taking into account that we detected seven Gamma–Omicron BA.1 co-infections while detected no Delta–Omicron BA.1 co-infection (which is more likely considering the dynamics of these VOCs in Moscow [21]), the second option is more plausible. It is likely that the mutation, which is characteristic of the Gamma variant, became established in the local lineage of the Omicron B.1.1.529/BA.1. Nevertheless, it is puzzling why this mutation has not occurred in Omicron BA.2. This observation is unusual and warrants further investigation.

Limitations of the developed PCR system include the fact that it can only detect one mutation for each variant. However, these selected mutations are present in a large number of sequenced genomes and have a low probability of disappearing spontaneously, making the system relatively robust. Additionally, our system is unable to distinguish between modern coronavirus variants Omicron JN.1 and Omicron KS.1 since it is impossible to design a reliable probe-based PCR set due to the high similarity between these variants’ genome sequences. Another limitation arises from the fact that the system was developed mainly for the retrospective analysis of samples isolated from humans infected with SARS-CoV-2 VOCs. While these samples are unlikely to be contaminated with related human coronaviruses, there is still such a possibility. Although we checked in silico for the absence of identified VOC-specific mutations in genomes of related human coronaviruses, we were unable to test the specificity experimentally. Therefore, if the system is to be used for purposes where the related human and/or bat coronaviruses may be present, it will need to be tested for cross-reactivity.

## 4. Materials and Methods

### 4.1. Search for the SARS-CoV-2 Mutant Variants’ Representative Genomes and Identification of the Variant-Specific Mutations

To identify variant-specific mutations that are unique to the analyzed SARS-CoV-2 VOC variants, genome sequences from the NCBI Virus online database (https://www.ncbi.nlm.nih.gov/labs/virus/vssi/#/, accessed on 30 September 2024) were extracted. Correspondence between SARS-CoV-2 lineages and VOC designations were taken from the official WHO website (https://www.who.int/publications/m/item/historical-working-definitions-and-primary-actions-for-sars-cov-2-variants, accessed on 30 September 2024) and GISAID platform (https://gisaid.org/hcov19-variants/, accessed on 30 September 2024). The sequence of the wild-type SARS-CoV-2 virus Wuhan-Hu-1 (NCBI access number: MN908947) was selected as the reference genome, relative to which the search for characteristic mutations was carried out. Accordingly, within the study, genome sequences were analyzed: for variant Alpha—100 genomes of lineages B.1.1.7 and Q; for variant Beta—100 genomes of lineage B.1.351; for variant Gamma—100 genomes of lineages and sublineages P.1; for variant Delta—115 genomes of lineages B.1.617.2 and AY; for Omicron variant—120 genomes of lineages B.1.1.529, BA.1, and BA.2; for modern sublineages of Omicron BA.2—200 genomes of lineages JN.1 and KS.1. NCBI accession numbers of all analyzed genome sequences are listed in Appendix A.

Among the analyzed genome sequences, a representative genome for each variant that contained all the characteristic mutations of the particular variant was identified. For all variants other than Omicron, a single representative genome was selected: Alpha—LC654482; Beta—MZ281047; Gamma—ON575495; Delta—ON324312 (here and below NCBI access numbers are provided). In the case of Omicron, the pattern of mutations in the BA.2 variant was different from that in BA.1 and B.1.1.529. Therefore, two representative genomes for the Omicron variant were selected, which characterize the two patterns of characteristic mutations detected: ON679744 for the BA.1/B.1.1.529 lineages and ON821528 for the BA.2 lineage. Additionally, two genomes, PP864229 and PQ212127, were selected as representative ones for Omicron BA.2 progeny variants JN.1 and KS.1, respectively. It is worth noting that genomic RNAs of these two Omicron BA.2 progeny variants share more than 99.94% identity and differ only in several single-nucleotide substitutions. The identified representative genomes were aligned with each other and with the reference wild-type genome of Wuhan-Hu-1. Based on a comparison of the representative genome sequences, specific mutations were selected to develop PCR sets (Figure 5):For the Alpha variant—nucleotide deletion 21991-21993 (del21991-21993; Y144del) in the glycoprotein S gene [22,23];For the Beta variant—nucleotide deletion 22281-22289 (del22281-22289; LAL242-244del) in the glycoprotein S gene [23];For the Gamma variant—samesense substitution of the AG nucleotides by TC at positions 28877-28878 (S202S) and GGG substitution by AAC at positions 28881-28883 (R203K-G204R) in the nucleocapsid protein N gene [24];For the Delta variant—the deletion of nucleotides 28248-28253 (del28248-28253; D119I-F120del) in the ORF8 gene [23];For the Omicron variant B.1.1.529/BA.1 lineages—deletion of nucleotides 21987-21995 (del21987-21995; GVY142-144del-Y145D) in the glycoprotein S gene [25,26];For the Omicron variant BA.2 lineage—deletion of nucleotides 21633-21641 (del21633-21641; LPP24-26del-A27S) in the glycoprotein S gene [27,28];For the Omicron’s variant BA.2 modern sublineages JN.1 and KS.1—substitution of the T by A at position 23005 (N481K), substitution of the G by A at position 23009 and nucleotide deletion 23010-23012 (E484K), and substitution of the TT by CC at positions 23018-23019 (F486P) in the glycoprotein S gene [29,30]. It should be noted that this mutation is common among both JN.1 and KS.1 and therefore allows them to be detected but not distinguished from each other.

### 4.2. Genomic RNA of SARS-CoV-2 Variants

Genomic RNA of the wild-type SARS-CoV-2 variant Wuhan (B.1.1.1, hCoV-19/Russia/Moscow_PMVL-1/2020), as well as of the mutant variants Alpha (B.1.1.7, hCoV-19/Netherlands/NoordHolland_20432/2020), Beta (B.1.351, hCoV-19/Russia/SPE-RII-27029S/2021), Gamma (B.1.1.28/P.1, hCoV-19/Netherlands/NoordHolland_10915/2021), Delta (B.1.617.2, hCoV-19/Russia/SPE-RII-32758S/2021), Omicron BA.1 (B.1.1.529 BA.1, hCoV-19/Russia/MOW-Moscow_PMVL-O16/2021), Omicron BA.2 (B.1.1.529 BA.2, hCoV-19/Russia/MOW-Moscow_PMVL-SN402/2022), Omicron BA.2 sublineage JN.1 (JN.1, hCoV-19/Russia/MOW-PMVL-LSCV-LD134/2023), and Omicron BA.2 sublineage KS.1 (KS.1, hCoV-19/Russia/SPE-RII-MH183935S/2024) was isolated from the cryopreserved stocks of the Russian State Collection of Viruses of the N.F. Gamaleya National Research Center of the Ministry of Health of Russia. Aliquots of viral stocks were lysed using 500 μL of the TRIzol reagent (Thermo Fisher Scientific, Waltham, MA, USA). Then, 100 μL of chloroform was added, and the mixture was centrifuged for 15 min at 4 °C and 13,400× *g*. The supernatant was diluted with 250 μL of isopropanol and incubated for 20 min at −20 °C. Next, the mixture was centrifuged again for 10 min at 4 °C at 13,400× *g*. RNA precipitate was washed with 500 μL of 75% ethanol and air-dried for 7 min. Then, the precipitated RNA was dissolved in 50 μL of TE buffer (10 mM Tris, 1 mM EDTA, pH 8.0) and heated at 65 °C for 10 min. RNA concentration was estimated using a NanoDrop 2000 Spectrophotometer (Thermo Fisher Scientific, Waltham, MA, USA) by measuring the optical density of the solution at 260 nm following the manufacturer’s standard protocol. Aliquots were stored at −70 °C.

### 4.3. Cryobank of Human RNA Isolated from Nasopharyngeal Swabs

Our laboratory staff collected a cryobank of total RNA extracted from human nasopharyngeal swabs collected between May 2021 and 30 September 2024. RNA was isolated using the RIBO-prep kit (AmpliSens, Moscow, Russia) according to the manufacturer’s standard protocol, digested with DNase I (Silex, Moscow, Russia), re-isolated using the RIBO-prep kit, and stored at −70 °C. From this cryobank, we randomly selected 52 samples, 11 from healthy donors and 41 from infected individuals whose infection had been clinically established and confirmed using a previously published SARS-CoV-2 test [31].

### 4.4. Primers and Probes Design

Sets of primers and probes were designed to specifically amplify the variant-specific mutations identified above in accordance with standard recommendations [32]. The absence of non-specific amplification of the human genome and transcriptome was verified using the Primer-BLAST online platform (https://www.ncbi.nlm.nih.gov/tools/primer-blast/, accessed on 30 September 2024) and also confirmed experimentally using total RNA isolated from nasopharyngeal swabs from healthy donors of different sexes from our laboratory personnel. Additionally, using Primer-BLAST, we checked the absence of non-specific amplification of the genomes of the related coronaviruses: SARS-CoV (NC_004718.3), MERS-CoV (NC_019843.3), and so-called human “common cold” coronaviruses—HCoV-OC43 (NC_006213.1), HCoV-HKU1 (NC_006577.2), HCoV-229E (NC_002645.1), HCoV-NL63 (NC_005831.2). The optimal annealing temperature for each primer/probe set was experimentally determined. And since the primers were designed with similar melting temperatures (~60 °C), their optimal annealing temperature was found to be 58 °C.

All oligonucleotides were synthesized commercially (DNA synthesis, Moscow, Russia). Their sequences are listed in Table 2.

### 4.5. Real-Time Polymerase Chain Reaction with Reverse Transcription

RT-PCR was performed using the C1000 instrument with the CFX96 fluorescent detection module (Bio-Rad, Hercules, CA, USA). The reaction mixture, 20 μL in volume, contained 5 μL of RNase-free deionized water, 5 μL of primers/probe mix (with a final concentration of 400 nM for each primer and probe), 5 μL of OneTube RT-PCR mix (Evrogen, Moscow, Russia) for reverse transcription and subsequent amplification, and 5 μL of RNA template. The PCR program consisted of 20 min at 48 °C for the reverse transcription, then 2 min at 95 °C, followed by 42 cycles of 15 s at 95 °C, 18 s at 58 °C and 20 s at 72 °C.

### 4.6. Data Analysis

Oligo Analyzer software version 1.0.3 (OligoSoftware, Kuopio, Finland) was used to design primers and probes. PCR data were analyzed using CFX Manager Software version 3.1 (Bio-Rad, USA). In silico specificity of the designed PCR sets and the absence of a non-specific amplification of human genome/transcriptome was analyzed using the online platform Primer-BLAST (https://www.ncbi.nlm.nih.gov/tools/primer-blast/, accessed on 30 September 2024), utilizing NCBI RefSeq Reference and Representative genomes for *Homo Sapiens* databases.

## 5. Conclusions

Taken together, in the current study, we have developed a multiplexed PCR system, which allows simultaneous detection of six VOC and two modern SARS-CoV-2 variants (Alpha, Beta, Gamma, Delta, Omicron B.1.1.529/BA.1, Omicron BA.2, and Omicron BA.2 progeny variants JN.1 and KS.1) by assessing three PCR reactions. Regarding the significant cost of high-throughput sequencing technologies, our system provides a cheaper alternative for scientific purposes, such as retrospective screening of SARS-CoV-2 variants to trace evolutionary patterns of the coronavirus. Additionally, since our system includes human mRNA detection as an internal control, it allows us to avoid false negative results in samples that have been stored for a long time and in which RNA might have been degraded. Moreover, it can be used to quantify the viral load in a sample in relation to human mRNA levels. This information can be useful for the investigation of correlations between viral loads and available clinical parameters of patients, the course of the disease, and its outcomes. However, the usefulness of this approach needs to be investigated separately. Furthermore, our system can be improved by adding new primers and probes that would allow the detection of newly emerging VOCs. This will make it possible to keep up with the rapidly evolving nature of the virus. Concerning the application of the system for IVD, the PCR oligonucleotide set for Omicron JN.1 and Omicron KS.1 might be used separately in a singleplex format to detect these modern SARS-CoV-2 variants. However, its applicability has to be validated in a separate study using the IVD guidelines.

## Figures and Tables

**Figure 1 ijms-26-01786-f001:**
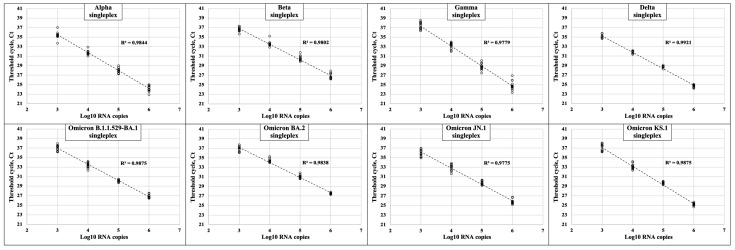
Sensitivity testing of the designed PCR sets. SARS-CoV-2 VOC variants Alpha, Beta, Gamma, Delta, Omicron B.1.1.529/BA.1, and Omicron BA.2, as well as modern Omicron BA.2 progeny variants JN.1 and KS.1, were tested with the corresponding singleplex PCR set using tenfold diluted standard solutions with 10^6^–10^2^ RNA genome copies per PCR reaction. Each standard solution was analyzed in 15 replicates. For the data obtained, a linear regression model was constructed, and the coefficient of linear determination (R^2^) is shown in each panel.

**Figure 2 ijms-26-01786-f002:**
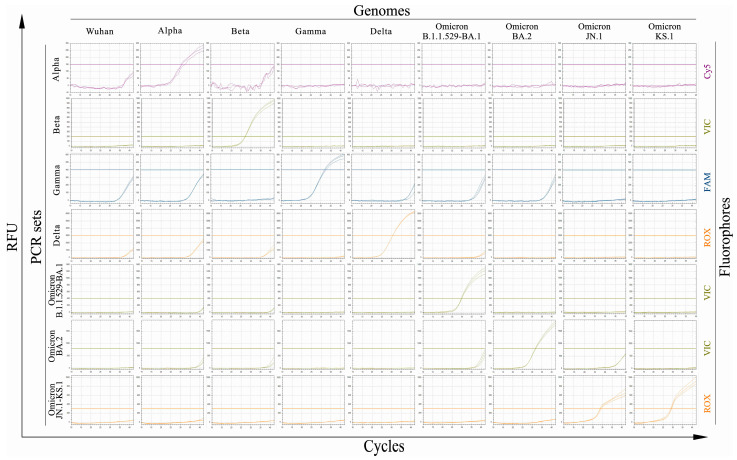
Specificity testing of the designed PCR sets. Oligonucleotide sets for detection of SARS-CoV-2 VOCs (Alpha, Beta, Gamma, Delta, Omicron B.1.1.529/BA.1, and Omicron BA.2), as well as of modern Omicron BA.2 progeny variants JN.1 and KS.1, were tested using 10^6^ RNA genome copies of each variant, including wild-type Wuhan, per reaction in triplicates. RFU, relative fluorescent units. FAM, VIC, ROX, and Cy5—fluorophores used in corresponding oligonucleotide probes.

**Figure 3 ijms-26-01786-f003:**
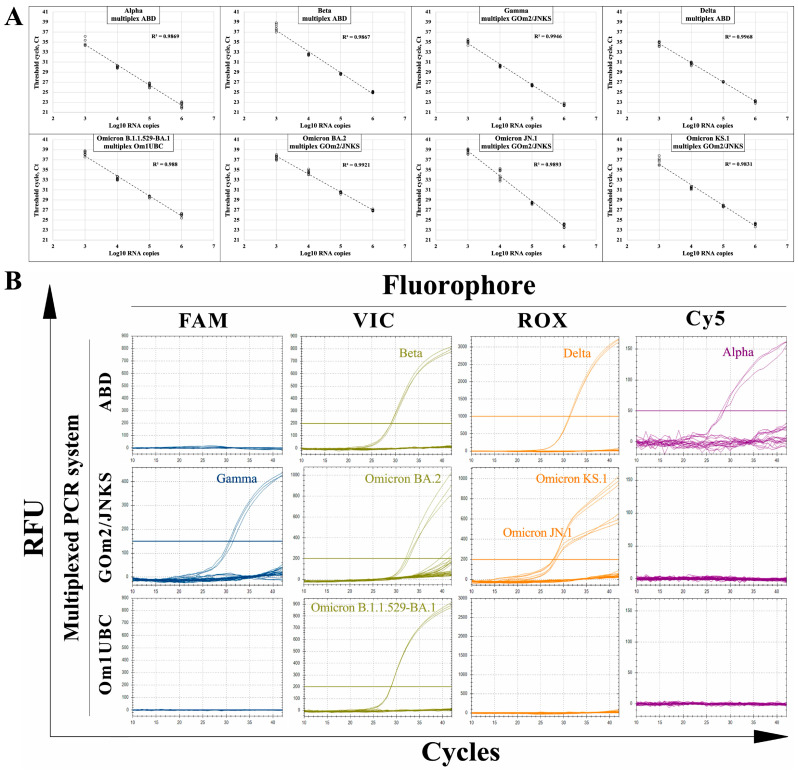
Sensitivity and specificity testing of the multiplex PCR systems. (**A**) SARS-CoV-2 VOC variants Alpha, Beta, Gamma, Delta, Omicron B.1.1.529/BA.1, and Omicron BA.2, as well as modern Omicron BA.2 progeny variants JN.1 and KS.1, were tested with the corresponding multiplexed PCR system using tenfold diluted standard solutions with 10^6^–10^2^ RNA genome copies per PCR reaction. Each standard solution was analyzed in 9 replicates. For the data obtained, a linear regression model was constructed, and the coefficient of linear determination (R^2^) is shown in each panel. (**B**) Each multiplexed PCR set was tested using 10^6^ RNA genome copies of each SARS-CoV-2 variant per reaction in triplicates. RFU, relative fluorescent units. Multiplexed PCR system, consisting of three combined PCR sets designated by their specificity: “ABD”—Alpha, Beta, and Delta; “GOm2/JNKS”—Gamma, parental Omicron BA.2, and modern Omicron BA.2 progeny variants JN.1 and KS.1; “Om1UBC”—Omicron B.1.1.529/BA.1 and human *UBC* gene. FAM, VIC, ROX, and Cy5—fluorophores used in corresponding oligonucleotide probes.

**Figure 4 ijms-26-01786-f004:**
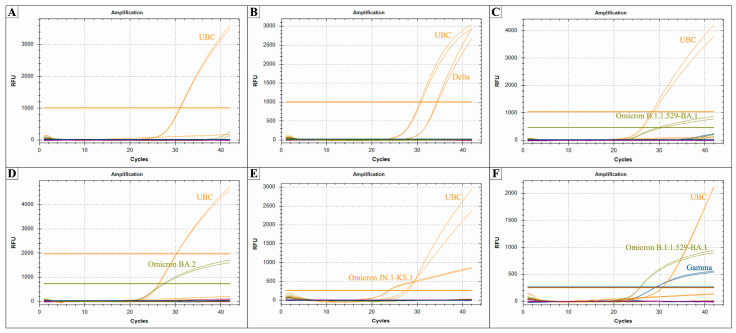
Testing of the multiplexed PCR system on human nasopharyngeal swabs. Representative results of the PCR analysis of a healthy donor (**A**), individual infected with Delta (**B**), Omicron B.1.1.529/BA.1 (**C**), Omicron BA.2 (**D**), Omicron BA.2 progeny variant JN.1 or KS.1 (**E**), and individuals containing simultaneously Omicron B.1.1.529/BA.1 and Gamma mutations (**F**) are shown. Each reaction was performed in duplicate. RFU, relative fluorescent units.

**Figure 5 ijms-26-01786-f005:**
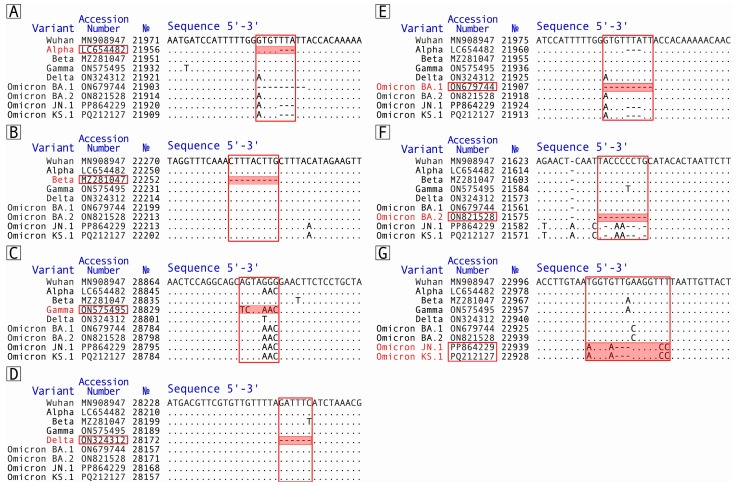
Variant-specific mutations of the SARS-CoV-2 VOCs. Sites with mutations unique for coronavirus variants Alpha (**A**), Beta (**B**), Gamma (**C**), Delta (**D**), Omicron BA.1/B.1.1.529 (**E**), Omicron BA.2 (**F**), and modern Omicron BA.2 progeny variants JN.1 and KS.1 (**G**) are shown. №, nucleotide’s number in the corresponding sequence. Figure was created using the online platform NCBI Basic Local Alignment Search Tool.

**Table 1 ijms-26-01786-t001:** PCR analysis of total RNA isolated from human nasopharyngeal swabs.

SARS-CoV-2 Variant Detected	Samples (N)	Ct *	Ct *UBC* *
Delta	5	27.7 (27.2–28.3)	31.2 (29.9–31.8)
Omicron B.1.1.529/BA.1	5	27.8 (25.1–28.5)	27.6 (27.2–32.0)
Omicron BA.2	19	25.3 (22.6–28.1)	27.8 (24.6–31.1)
Omicron JN.1/KS.1	5	25.3 (24.1–29.7)	30.1 (29.4–30.7)
Gamma+Omicron B.1.1.529/BA.1	7	Gamma: 27.1 (22.6–27.7)	29.7 (25.5–30.2)
		Omicron: 24.7 (23.3–28.3)	
None	11	–	29.0 (26.3–31.6)

*—values are presented as median (interquartile range) for all samples where the corresponding SARS-CoV-2 variant is detected.

**Table 2 ijms-26-01786-t002:** Oligonucleotides used as variant-specific PCR sets.

Oligonucleotide Designation	Sequence (5′-3′)	Localization in Corresponding Genome (5′-3′) *	PCR Fragment Size
Alpha_F	AATGATCCATTTTTGGGTGTTTACC	21956-21980	138
Alpha_R	CTGTTTTCCTTCAAGGTCCATAAG	22093-22070	
Alpha_probe	Cy5-GGAAAGTGAGTTCAGAGTTTATTCTAGT-BHQ2	22003-22030	
Beta_F	AGATTTGCCAATAGGTATTAACATC	22223-222472	89
Beta_R	CTGTCCAACCTGAAGAAGAATC	22311-22290	
Beta_probe	VIC-CTAGGTTTCAAACTTTACATAGAAGTT-BHQ2	22249-22275	
Gamma_F	CAAGGAACAACATTGCCAAAAGG	28725-28747	140
Gamma_R	CTAGCAGGAGAAGTTCGTTTAGA	28864-28842	
Gamma_probe	FAM-GTTGCGACTACGTGATGAGGAACG-BHQ1	28814-28791	
Delta_F	CATGACGTTCGTGTTGTTTTAATC	28171-28194	140
Delta_R	CACTGCGTTCTCCATTCTGG	28310-28291	
Delta_probe	ROX-CCAGTTGAATCTGAGGGTCCACCA-BHQ2	28261-28284	
OmicronBA.1_F	TGTAATGATCCATTTTTGGACCAC	21900-21923	105
OmicronBA.1_R	CTGAGAGACATATTCAAAAGTGCA	22004-21981	
OmicronBA.1_probe	VIC-GGAAAGTGAGTTCAGAGTTTATTCTAGT-BHQ2	21944-21971	
OmicronBA.2_F	TAATCTTATAACCAGAACTCAATCATA	21562-21588	147
OmicronBA.2_R	ATAGCATGGAACCAAGTAACATTG	21708-21685	
OmicronBA.2_probe	VIC-TACCCTGACAAAGTTTTCAGATCCTC-BHQ2	21617-21642	
JN/KS_F	TTACAGGCTGCGTTATAGCTTG	22791-22812	183
JN/KS_R	GAAAGTAACAATTAGGACCTTTACCTT	22973-22947	
JN/KS_probe	ROX-GATCGCTTAGGAAGTCTAAACTCAAACC-BHQ2	22866-22893	
UBC_F	TTGGGTCGCAGTTCTTGTTTG	22-42	131
UBC_R	TGCCTTGACATTCTCGATGGT	152-132	
UBC_probe	ROX-TCGCTGTGATCGTCACTTGACAATG-BHQ2	47-71	

* NCBI accession numbers of used genome sequencies: Alpha—LC654482; Beta—MZ281047; Gamma—ON575495; Delta—ON324312; Omicron BA.1/B.1.1.529—ON679744; Omicron BA.2—ON821528; Omicron JN.1—PP864229; Omicron KS.1—PQ212127. For the oligonucleotide probes, designing fluorophores (FAM, VIC, ROX, and Cy5) and corresponding quenchers (BHQ1/2, Black Hole Quencher 1/2) were used.

## Data Availability

The data presented in this study are available in the article and Appendix A.

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
