# Peer review of "Real-Time PCR-Based Test as a Research Tool for the Retrospective Detection and Identification of SARS-CoV-2 Variants of Concern in a Sample"

_ijms, 2025, doi:10.3390/ijms26051786_

Round 1

Reviewer 1 Report

Comments and Suggestions for Authors

The authors developed a multiplex real-time PCR-based system for the rapid detection of multiple mutant strains of SARS-CoV-2 VOCs, including Alpha, Beta, Gamma, Delta, Omicron B.1.1.529/BA.1, BA.2, and their progeny JN.1 and KS.1. By analyzing variant-specific mutations, they designed primers and probes targeting different variants, verifying their sensitivity and specificity (no cross-reactivity) in both single- and multiplex PCR assays. The system incorporates the human ubiquitin C gene as an internal reference and was successfully applied to 52 nasopharyngeal swab samples (including cryopreserved specimens) to differentiate between mixed infections of mutant strains. Given its methodological soundness and potential clinical relevance, I support its publication in the International Journal of Molecular Sciences at this stage. However, I have the following key comments for further improvement:

1. The use of a paper-based multiplex real-time PCR system for SARS-CoV-2 variant identification has been previously reported. Please refer to the following papers, such as Xueting Lu et al., Trends in Analytical Chemistry, 2025, 184, 118-131, and Udugama B et al., ACS Nano, 2020, 14(4):3822-3835. The authors should clarify how their approach advances beyond existing methods to enhance novelty.

2. The abstract and introduction should explicitly state which PCR-based methodological principles (e.g., allele-specific PCR, probe-based detection, or melt-curve analysis) were employed to detect different SARS-CoV-2 variants.

3. Given that JN.1 and KS.1 are descendants of Omicron and share many protein mutations, how does the designed PCR system successfully distinguish between these closely related subvariants? The manuscript should provide a detailed explanation or experimental validation supporting this differentiation.

4. Figure 2 is unclear in the manuscript. The authors should provide a higher-resolution image to enhance readability and interpretation.

Author Response

First of all, together with co-authors we would like to thank the Reviewer for the time they have devoted to thorough examination of our work and their valuable comments. Point-to-point response to the Reviewer’s comments is presented below.

Response to Reviewer 1 comments:

The authors developed a multiplex real-time PCR-based system for the rapid detection of multiple mutant strains of SARS-CoV-2 VOCs, including Alpha, Beta, Gamma, Delta, Omicron B.1.1.529/BA.1, BA.2, and their progeny JN.1 and KS.1. By analyzing variant-specific mutations, they designed primers and probes targeting different variants, verifying their sensitivity and specificity (no cross-reactivity) in both single- and multiplex PCR assays. The system incorporates the human ubiquitin C gene as an internal reference and was successfully applied to 52 nasopharyngeal swab samples (including cryopreserved specimens) to differentiate between mixed infections of mutant strains. Given its methodological soundness and potential clinical relevance, I support its publication in the International Journal of Molecular Sciences at this stage. However, I have the following key comments for further improvement:

Point 1. «The use of a paper-based multiplex real-time PCR system for SARS-CoV-2 variant identification has been previously reported. Please refer to the following papers, such as Xueting Lu et al., Trends in Analytical Chemistry, 2025, 184, 118-131, and Udugama B et al., ACS Nano, 2020, 14(4):3822-3835. The authors should clarify how their approach advances beyond existing methods to enhance novelty.»

Response 1: The reviews by Xueting Lu et al. and Udugama B et al., which we referred to according to the Reviewer’s advice, comprehensively discuss the approaches and challenges of developing COVID-19 in vitro diagnostic (IVD) tools. In these papers the authors indicate that among the available IVD methods for detecting SARS-CoV-2 infection, nucleic acid testing has become the primary method due to a number of advantages. Nevertheless, the virus constantly evolves and its genomic RNA accumulates mutations, that poses a challenge for the development of nucleic acid-based IVD assays. These assays have to be constantly updated and validated to be able to detect new emerging variants. However, since co-infection with different virus variants is a highly rare event, as well as emerging SARS-CoV-2 variants are tended to replace previous ones from the population, IVD tests are commonly designed to detect only those variants that currently dominate worldwide. In contrast to IVD tests, SARS-CoV-2 detection assays for research purposes should not only detect the presence of viral RNA in a sample, but also identify a specific variant. And in case of retrospective analysis, even those variants that are not currently circulating. In this context, the multiplexed PCR system suggested in our manuscript allows detecting all SARS-CoV-2 VOCs identified during COVID-19 pandemic, regardless of their temporal relationships. The system provides a cheaper alternative to sequencing approaces, e.g., for retrospective screening of SARS-CoV-2 variants. Additionally, since our system includes human mRNA detection as an internal control, it allows to avoid false negative results in samples that have been stored for a long time and in which RNA might have been degraded. Thus, we believe that our system is a useful tool for scientific research rather than for IVD of COVID-19. In the amended manuscript we emphasized this moment in the article title, as well as in the Discussion (page 6, lines 149-170) and Conclusions sections (page 11, lines 352-367).

Point 2. The abstract and introduction should explicitly state which PCR-based methodological principles (e.g., allele-specific PCR, probe-based detection, or melt-curve analysis) were employed to detect different SARS-CoV-2 variants.

Response 2: The system developed uses probe-based detection technique to detect different SARS-CoV-2 variants. According to the Reviewer’s comment we added this information into the Abstract (page 1, line 25) and Introduction sections (page 2, line 63).

Point 3. Given that JN.1 and KS.1 are descendants of Omicron and share many protein mutations, how does the designed PCR system successfully distinguish between these closely related subvariants? The manuscript should provide a detailed explanation or experimental validation supporting this differentiation.

Response 3: Our system can detect both Omicron JN.1 and KS.1 subvariants, but is not able to distinguish among them. Their genomic RNAs share more than 99,94% identity and differs only in several single nucleotide substitutions. Therefore, it is impossible to design primers/probes for their robust distinguishing. We apologize for the inconvenience with this issue. In the amended manuscript we clarified this issue in the Materials and Methods section (page 8, lines 252-254 and lines 273-275).

Point 4. Figure 2 is unclear in the manuscript. The authors should provide a higher-resolution image to enhance readability and interpretation.

Response 4: The manuscript was prepared using Word template according to the Journal rules. Unfortunately, Word automatically compresses the inserted images, so the resolution of figures might be lost. To avoid this, we generated the TIFF images of all figures with a high resolution and attached them to the amended manuscript as separate files.

Reviewer 2 Report

Comments and Suggestions for Authors

The manuscript submitted for review “A real-time PCR based test for the retrospective detection and identification of SARS-CoV-2 variants of concern in a sample” by Makarova et al. describes the development of Real Time RT PCR for the detection of the 7th variant of SARS-CoV-2. The authors have approached the development of the protocol/method consistently and professionally.

Regarding the result - Percent match: 26%, related to the iThenticate report, after reviewing the result obtained, my conclusion is that it is cumulative from matches of individual words and phrases, which are universal and often used in professional terminology or cannot be defined as plagiarism.

However, there are some gaps related to the validation of the protocol and the description of the experiments.

Other viruses - coronaviruses and related viruses - should be included in the study to determine the specificity of the method.

To determine the sensitivity of the method, it is necessary to conduct tests with dilution of the obtained RNA (not the initial sample, because the amount of the obtained RNA depends on the method and the experience of the performer). As an example of the above two, I attach the following manuscript: https://doi.org/10.3390/vetsci9060272

The method should be applied with an IVD method, which should be used as a gold standard for calculating specificity/sensitivity (for example: https://doi.org/10.3390/microorganisms12010180 ).

The results mention the study of clinical samples - nasopharyngeal. There is no data in the material and methods section on the number of samples, origin, etc. information about them is available. There should be an inventory of positive and negative samples, the method by which they were examined. It would be nice to use the same method for comparison and assessment of sp/sensitivity.

The authors have to determine, after the RNA dilutions, at the corresponding concentrations, at which cycle the curve appears.

When conducting a multiplex reaction, how will the curves be distinguished in simultaneously positive samples for Beta and Omicron variants, given that probes with the same dyes are used - VIC - BHQ2?

In Table 2, add the localization of the primers in the virus genome, the gene being tested and the size of the fragment.

There is a lot of information in the world literature that will help improve the discussion, which needs to be deepened and expanded.

Author Response

First of all, together with co-authors we would like to thank the Reviewer for the time they have devoted to thorough examination of our work and their valuable comments. Point-to-point response to the Reviewer’s comments is presented below.

Response to Reviewer 2 comments:

The manuscript submitted for review “A real-time PCR based test for the retrospective detection and identification of SARS-CoV-2 variants of concern in a sample” by Makarova et al. describes the development of Real Time RT PCR for the detection of the 7th variant of SARS-CoV-2. The authors have approached the development of the protocol/method consistently and professionally.

Regarding the result - Percent match: 26%, related to the iThenticate report, after reviewing the result obtained, my conclusion is that it is cumulative from matches of individual words and phrases, which are universal and often used in professional terminology or cannot be defined as plagiarism.

However, there are some gaps related to the validation of the protocol and the description of the experiments.

Point 1: Other viruses - coronaviruses and related viruses - should be included in the study to determine the specificity of the method.

Response 1: First of all, we would like to highlight that the PCR system developed in our study is a tool for scientific research rather than for in vitro diagnostic of COVID-19. The system provides a cheaper alternative to sequencing approaches, and ca be used, for example, for retrospective screening of SARS-CoV-2 variants. Additionally, since our system includes human mRNA detection as an internal control, it allows to avoid false negative results in samples that have been stored for a long time and in which RNA might have been degraded. In the amended manuscript we emphasized this moment in the article title, as well as in the Discussion (page 6, lines 149-170) and Conclusions sections (page 11, lines 352-367). However, we agree with the Reviewer that the system still should be tested for specificity using related viruses. Unfortunately, it is impossible to perform testing the system experimentally within the time we have for revision. However, we performed in silico analysis of the specificity of our PCR system using alignment of the SARS-CoV-2 representative genomes (using Wuhan-Hu-1 as query) and genomes of the SARS-CoV (NC_004718.3), MERS-CoV (NC_019843.3) and so-called human “common cold” coronaviruses – HCoV-OC43 (NC_006213.1), HCoV-HKU1 (NC_006577.2), HCoV-229E (NC_002645.1), HCoV-NL63 (NC_005831.2). As a result, we detected very low similarity between genomic RNA sequencies of SARS-CoV-2 variants and other viruses, except for SARS-CoV that shares 80% identity (see the picture below).

Since only SARS-CoV shares a relatively high degree of similarity with SARS-CoV-2, we checked the identified variant-specific mutations for their specificity (see the picture below).

As a result, all variant-specific mutations were not presented in SARS-CoV genome. Additionally, we tested all three multiplexes consisting the system for specificity using NCBI Primer-BLAST online resource and genome sequencies of SARS-CoV, MERS-CoV and common cold coronaviruses. No non-specific amplification was detected. Thus, we believe that experimental testing would provide the same results. We mentioned this issue in the Materials and Methods section (page 10, lines 316-320).

Point 2: To determine the sensitivity of the method, it is necessary to conduct tests with dilution of the obtained RNA (not the initial sample, because the amount of the obtained RNA depends on the method and the experience of the performer). As an example of the above two, I attach the following manuscript: https://doi.org/10.3390/vetsci9060272

Response 2: Within the framework of the current study, we used cryopreserved samples of already isolated human total RNA, which have been collected during COVID-19 pandemic since 2020. Only these samples contain coronavirus VOC variants that have been eliminated from the population. Unfortunately, these samples are very limited in volume and we were not able to dilute them for testing the sensitivity of the primers/probe sets during the PCR system designing step. To avoid this problem, similar to the paper by Sirakov I et al. provided by the Reviewer, we used standard solutions of genomic RNA isolated from the viral stocks of different SARS-CoV-2 variants. Thus, we extracted genomic RNAs from the viruses, evaluated the concentration using NanoDrop 2000 device, diluted RNA to obtain standard solutions of 106-102 genomic RNA copies per PCR reaction, and used these solutions to test the sensitivity of both singleplex and multiplexed primers/probe sets. Using this approach, we determined the lower limit of detection (LLOD) for our PCR system being 103 genomic RNA copies per reaction. Indeed, the amount of the obtained RNA depends on the method and the experience of the performer; however, RNA isolation methods were out of the scope of our study. According to the results we obtained, the performer should obtain at least 3·103 genomic RNA copies (one PCR reaction per three multiplexes with given LLOD) irrespective of the particular extraction method to specify the SARS-CoV-2 VOC variant in a sample using our system.

Point 3: The method should be applied with an IVD method, which should be used as a gold standard for calculating specificity/sensitivity (for example: https://doi.org/10.3390/microorganisms12010180).

Response 3: Unfortunately, since in the current study we developed PCR system for the detection of SARS-CoV-2 variants that have been already cleared from the population, there is no IVD methods available on a market for detection of all VOC variants tested by us. Additionally, as we mentioned above, the system we created is a tool for scientific research rather than for IVD of COVID-19. Thus, the comparison of our PCR system with developed IVD methods was beyond the scope of the study.

Point 4: The results mention the study of clinical samples - nasopharyngeal. There is no data in the material and methods section on the number of samples, origin, etc. information about them is available. There should be an inventory of positive and negative samples, the method by which they were examined. It would be nice to use the same method for comparison and assessment of sp/sensitivity.

Response 4: According to the Reviewer’s comment, we transferred the information concerning samples of human RNA isolated from nasopharyngeal swabs from the “Results” section into the “Materials and Methods” section (page 10, lines 302-309).

Point 5: The authors have to determine, after the RNA dilutions, at the corresponding concentrations, at which cycle the curve appears.

Response 5: The information concerning the threshold cycle (Ct) for standard genomic RNA solutions and human RNA from nasopharyngeal swabs can be found on Figures 1 and 3A, as well as in Table 1 and Supplementary Table S1. Concerning the dilution of human RNA from nasopharyngeal swabs, as we mentioned above, the volume of these samples, unfortunately, was too low to perform such analysis.

Point 6: When conducting a multiplex reaction, how will the curves be distinguished in simultaneously positive samples for Beta and Omicron variants, given that probes with the same dyes are used - VIC - BHQ2?

Response 6: Indeed, PCR primer/probe sets for Beta and Omicron variants use the same dye - VIC. However, these sets are combined in different multiplexes (ABD and Om1UBC, respectively) and each multiplex should be performed and analyzed in a separate PCR reaction, so the dyes are not interfering. We apologize for the inconvenience with this issue. In order to avoid the confusion of readers, in the amended manuscript we clarified this issue (page 2, line 75; page 4, lines 108-109; page 5, lines 131-132; page 11, line 352).

Point 7: In Table 2, add the localization of the primers in the virus genome, the gene being tested and the size of the fragment.

Response 7: In accordance with the Reviewer’s comment, we added the information concerning localization of the primers in the virus genome and size of the PCR fragment in Table 2, as well as indicated the gene being tested in the Materials and Methods section (page 8, lines 257-275).

Point 8: There is a lot of information in the world literature that will help improve the discussion, which needs to be deepened and expanded.

Response 8: In accordance with the Reviewer’s comment, we deepened and expanded the Discussion section (page 6, lines 149-170; page 7, lines 201-214).

Round 2

Reviewer 1 Report

Comments and Suggestions for Authors

No more comments.

Author Response

We thank the Reviewer for the positive evaluation of our work and the help with its improvement.

Reviewer 2 Report

Comments and Suggestions for Authors

The manuscript submitted for review after the first review with a title “Real-time PCR based test as a research tool for the retrospective detection and identification of SARS-CoV-2 variants of concern in a sample” was improved by the authors, according to recommendations from some of the reviewers. The authors' responses to my recommendations are reasoned, detailed, and honest. It is a credit to the authors that they approached it honestly and respectfully, because some authors allow themselves to write results without having conducted research. In this case, the authors think that there is no need for specificity and sensitivity reactions, because the protocol they developed is for scientific purposes and the specificity of the primers has been checked by software.

In this regard, I would like to clarify that sometimes theory diverges from practice, because PCR involves different components, as well as nucleic acids, and there are different reaction parameters, for example: for scientific purposes I isolate a virus from bats, which I expect to be SARS-CoV-2, but it is not (but is closely related) and in this case I get a false-positive result. This is just an example.

My opinion - based on a lot of experience with different viruses - differs from that of the authors, how this study should be structured and what analyses should be conducted. Any method for a scientific or diagnostic purpose must have a certain specificity and sensitivity. Sensitivity and specificity are also used in science, this is not only for diagnostics. On the other hand, conducting studies that we know how they will end and spending money on this is unacceptable, but there are things that are mandatory.

My opinion differs from that of the authors, but a different opinion cannot be a reason for me as a reviewer to reject their point of view. My opinion is that for this type of journal (high IF) the analyses should be more diverse, also, as I pointed out in the first review - there are many studies and literature, which allows for a deep and extensive analysis and discussion (and this may cover the lack of some analyses mentioned above).

I will point out "minor revisions" related to what I wrote above.

If the Editors decide, they can publish it in this form or reject it. My vision of the experimental design is different from that of the authors, but this is an assessment for the Editors and readers (my task is to help better present the ideas, results and analysis, and to preserve the reputation of the journal).

Author Response

Once again, we would like to thank the Reviewer for the comprehensive analysis of our study. First of all, we would like to emphasize that we agree with the Reviewer concerning the sensitivity and specificity issues. Since we are not native English speakers, we suppose that our previous response has been interpreted by the Reviewer as “there is no need for specificity and sensitivity reactions for scientific purposes”, which is not true. We apologize for the possible misunderstanding. We sure that these issues are of a great importance, so we performed independent biological experiments with replicates to test the specificity and sensitivity of our system using RNAs from SARS-CoV-2 viral stocks as the only source of the variants that had been eliminated from the circulation. Our system has been developed and is applicable for the specific purpose – retrospective analysis of samples, isolated from humans infected with SARS-CoV-2 VOCs. In this regard, these samples are unlikely to be contaminated with bat or other human coronaviruses. And although we checked in silico the absence of VOC-specific mutations in related human coronaviruses’ genomes, we agree with the Reviewer that sometimes theory diverges from practice, unfortunately, quite often. Therefore, we pointed out the issue concerning possible cross-reactivity of our system with other viruses as a limitation of our study (page 7, lines 215-228). Nevertheless, despite all limitations of the system mentioned in the text, we believe that we showed its performance and applicability for the stated purpose, and the results of the study may be of interest for the readers of the International Journal of Molecular Sciences. The system, if not applied as it is, might be adapted for the particular aims by the researches.